# Structural Equation Model for Developing Person-Centered Care Competency among Senior Nursing Students

**DOI:** 10.3390/ijerph181910421

**Published:** 2021-10-03

**Authors:** Ji-Yeong Yun, In-Young Cho

**Affiliations:** 1Department of Nursing, Jesus University, 383 Seowon-ro, Wansangu, Jeonju-si, Jeollabukdo 54989, Korea; lilyjy@jesus.ac.kr; 2College of Nursing, Chonnam National University, 160 Baekseo-ro, Dong-gu, Kwang-ju 61469, Korea

**Keywords:** clinical practicum adaptation, self-awareness, nursing students, nursing professionalism, person-centered care competency

## Abstract

Recent health care developments have emphasized person-centered care, which highlights individualized treatments rather than focusing solely on the nature of a given disease. Thus, we aim to identify the factors and construct a structural equation model for developing person-centered care competency among senior nursing students based on the social cognitive career theory and a subsequent literature review. We use a hypothetical model to examine the factors influencing person-centered care competency, and using a structured questionnaire, and we collect data on self-awareness, the clinical learning environment, clinical practicum adaptation, nursing professionalism, empathy, and person-centered care competency. The participants include 383 third- and fourth-year senior nursing students who had undergone at least one semester of clinical practice in South Korea. SPSS/WIN 26.0 is used to analyze all obtained data, while AMOS 25.0 is used for structural equation modeling. The final model is confirmed to be suitable for explaining and predicting person-centered care competency among participants. Nursing professionalism, empathy, clinical practicum adaptation, self-awareness, and the clinical learning environment explained 38.8% of the total variance among participants. Strategies and interventions designed to enhance person-centered care competency for senior nursing students should particularly focus on nursing professionalism, empathy, clinical practicum adaptation, self-awareness, and the clinical learning environment.

## 1. Introduction

Human care is a central feature of the nursing profession [1]. Given that patients play a major role in deciding their treatment and care, they increasingly demand high-quality person-centered care (PCC)—the provision of individualized care rather than solely focusing on disease treatment [2,3,4]. PCC aims to help patients live meaningful lives through holistic methods considering entire livelihoods, including individual preferences and needs [5]. According to Park and Choi [6], the word “patient” tends to reduce an individual to someone who simply receives medical services; thus, when providing PCC, it is important to recognize the individual as a whole person who is actively involved in their care, as well as consider each patient’s physical, social, mental, and spiritual aspects [7].

Due to its increasing popularity globally, the Institute of Medicine [8] suggests that all care professionals prioritize PCC by establishing it beyond the hospital environment (e.g., in multidisciplinary work and evidence-based practice). Western countries, such as Norway, the United Kingdom, and the United States, have even designated PCC as a part of their respective national programs; thus, ensuring its adoption in regular nursing curricula. Efforts are, thus, directed at providing health delivery systems focused on individualized human care [4,7].

Many previous studies have also reported positive outcomes related to PCC. These include improved self-management [9], satisfaction and feelings of well-being, improved physical functioning, and shortened hospitalization periods for patients [10,11,12]. For nurses, related factors and outcomes include increased compassion satisfaction, improved interpersonal relationships, and reduced job stress [3,13].

Based on the above, given that PCC is a highly important way to provide holistic care, appropriate nursing competency (person-centered care competency: PCCC) to implement humanistic care is the most significant factor for ensuring proper PCC [6]. Leininger [14] previously identified nurses as unique care providers in the “caring experience,” as they promote healing while improving the health outcomes of those who are well, unwell, disabled, and dying [1]. This makes nurses key contributors in the PCC context, as they are often responsible for providing 24 h direct care tailored to individual patient needs and values; this is ensured through developing high-quality therapeutic relationships while establishing trust [6].

In line with these changes worldwide, senior nursing students—who are the future nurses—should strive to adapt to the changing health care paradigm as expected of nurses [15]. Considering that PCCC can be strengthened through continuous learning and curricula [3], there have been recent efforts to discuss the establishment of an integrated curriculum and introduce it gradually in undergraduate curricula to ensure PCC practice in the future [4,6,16].

However, at the initial stage, there remains a limitation, in that current nursing education is centered on acute-stage nursing and core skills enhancement. Nevertheless, a curriculum focusing on individualized humanistic care and efforts to enhance PCCC in various clinical settings after graduation remains a significant task for future nursing education [15,16].

Previous studies stated that success is achieved through self-awareness, professional competencies/beliefs, and empathy, as well as enhancing the clinical environment [4,10,12]. In addition, important characteristics of the current nursing practice include enhancing nursing students’ ability to develop and continue therapeutic relationships with patients while having autonomy and control over the practice environment. This leads to PCCC, which can be provided through systematic nursing curricula that integrate elements of professional knowledge with empathy [10].

As many senior nursing students become key nurses in PCCC, educators must grasp the qualities required to implement PCC in a way that encompasses all human care beyond the hospital environment; PCCC programs should, thus, be implemented based on relevant nursing career-related factors in the undergraduate curricula offered at nursing colleges [6,12,17]. Given that nursing education can provide students with relevant knowledge, professional values, and appropriate attitudes, curricula should be updated to increase holistic and humanistic care concepts and a systematic strategy based on a multi-faceted analysis should be established [16].

With this importance on PCCC, thus far, many theoretical and clinical advances have been made through global research. Most of the studies used exploratory approaches to examine specific factors, such as self-awareness, empathy, active listening, and past clinical performance [6,10,13]. However, as PCCC is an integrated concept difficult to explain with only a single variable, along with the individual aspect, the environment and learning experience aspects should also be considered important. In addition, these competencies should also be developed over the long-term and educators require a model that systematizes all relevant factors; thus, highlighting the direct and indirect pathways.

Therefore, to construct a hypothetical model to reach the target behavior (PCCC), we intend to use the social cognitive career theory (SCCT), which proposes that individuals who are exposed to certain career environments later pursue relevant career interests and goals. It further emphasizes the interplay between several personal and contextual characteristics, learning experiences, and outcome expectation factors to reach goal behaviors for college students, thereby promoting the development of career-related goal behavior after graduation [17].

We, therefore, judge that the SCCT is an appropriate framework for guiding senior nursing students in gaining the PCC competencies required for clinical practice.

### 1.1. Aims and Hypotheses

We aim to test a model informed by the SCCT framework [17] to examine, in detail, the structural relationship between personal, contextual, and outcome expectation factors, as well as learning experiences associated with becoming a new nurse with PCCC among current senior nursing students; thus, facilitating pedagogical implementation. Our findings could provide guidance on how to support the design of feasible interventions and strategies for the development of PCC programs. Thus, the objectives of this study were: first, to verify the suitability of the PCCC model for senior nursing students based on the SCCT of Lent, Brown, and Hackett [17]; second, to clarify the direct and indirect effects of related factors on PCCC.

### 1.2. Conceptual Framework and Hypothetical Model

Using the SCCT framework by Lent, Brown, and Hackett [17], as well as existing literature, we develop a theoretical model for nursing students to be trained in PCCC. Lent, Brown, and Hackett’s [17] SCCT, a model developed parallel to the rise of positive psychology, is focused on human positive functioning. Recently, it has attracted attention, particularly because it enables a more comprehensive understanding and development of an individual’s complex career-related behavior. SCCT emphasizes that one’s beliefs and expectations about their ability to accomplish a certain task successfully, play a key role in their career behavior by influencing their interests, values, and abilities. This model emphasizes the concepts of personal factors, contextual factors, learning experiences, and expectations for an individual to reach a specific (goal behavior) as shown in Figure 1.

This theory hypothesizes that an individual’s career-related behavioral goal development is not a mere result of individual characteristics and the environment, but rather a result of interactions between the personal and contextual learning and expectation factors, and that these expectations are affected by personal, contextual, and learning experiences. Thus, according to this model, when individuals set goals anticipating positive outcomes, their confidence in these goals increases; furthermore, by actively taking action to achieve the goal, the likelihood of success is relatively increased.

In the current study, the variables were derived based on the SCCT framework and previous studies. The personal factor was treated as self-awareness, since nursing students must know themselves both cognitively and emotionally [18]. In addition, the contextual factor was treated as the clinical learning environment, which refers to a set of components that, together, influence the education outcomes of learners within the clinical settings [19]. Supportive environments have been shown to affect competence positively, thereby enhancing clinical care competency [20]. These factors are aligned, accordingly, with personal and contextual conditions, and, thus, are considered exogenous variables in this study.

Regarding the expectation concepts examined in this study, outcome expectations refer to an individual’s belief concerning the long-term consequences of a career [17]. During the formation of goal behavior, outcome expectations are partly determined since people often expect to achieve a desirable outcome in a career when they view themselves as capable performers [21]. Here, we consider the students’ expected outcome to become qualified professional nurses (Nursing professionalism), which emanates from a strong sense of self-belief that is capable of influencing them physically and psychologically [22]. In sum, nursing students often expect to achieve a desirable outcome in their careers when they view themselves as capable performers.

According to the SCCT, the development of expectations depends on the learning experience. In this study, learning experience is conceptualized as the nursing students’ clinical practicum adaptation, regarded as learning by adapting to clinical practice. In turn, the learning experience is affected by the influence of personal and contextual factors.

Therefore, these personal, contextual, learning experiences, and expectation factors will have a direct and indirect effect on PCCC (which is the target goal behavior). Additionally, following an extensive literature review, we add empathy as a mediating factor of PCCC [5,10]. Accordingly, we compose a PCCC hypothetical model by establishing that personal factors (self-awareness) and contextual factors (perceived clinical learning environment) could impact the outcome expectation factor (nursing professionalism) and empathy of nursing students mediated by learning experience (clinical practicum adaptation), as shown in Figure 2.

## 2. Materials and Methods

### 2.1. Design

This study used a structural equation model to suggest and verify a hypothetic model for the influential factors of PCCC among senior nursing students.

### 2.2. Participants

We collected data from third- and fourth-year nursing students who had undergone at least one semester of clinical practice in South Korea.

Comrey and Lee [23] stated that a sample size of approximately 300 is good for factor analysis; therefore, we considered a sample of 300 participants to be sufficient for our analysis. Based on this, the final sample consisted of 383 participants, which was considered sufficient.

### 2.3. Data Collection

We collected data after approval by the Bioethics Review Committee of the second author’s university (details omitted for double-blind reviewing). All procedures performed in this study followed the ethical standards of the institutional and/or national research committee. Data collection was conducted via an online survey (which was safer than engaging in face-to-face surveys due to the COVID-19 pandemic) between November and December 2020, in South Korea. The research announcement was promoted on cross-regional social media platforms where nursing students from all regional backgrounds share information and opinions about their studies and careers.

We provided respondents with sufficient information about the study purpose and all related procedures. We also assured participants regarding the anonymity and confidentiality of their responses and asserted that any collected data would be used for research purposes only. Participants were told that they could withdraw from the study at any time if desired. Those who (voluntarily) agreed to participate answered the questions. After the survey, the researcher sent mobile coupons to the participants as a token of appreciation for completing the survey.

### 2.4. Measurements

#### 2.4.1. Self-Awareness

We measured self-awareness using a scale developed by Fenigstein et al. [24] and translated into Korean by Eun [25]. Twenty items measured 3 sub-scales on a 5-point Likert scale, including private self-awareness, public self-awareness, and social anxiety. A higher score indicated higher-level self-awareness. For this study, a confirmatory factor analysis was performed to extract sub-factors. Reliability was Chonbach’s α = 0.80 for Fenigstein et al.’s [24] study, Cronbach’s α = 0.74 for Eun’s [25] study, and Cronbach’s α = 0.69 for this study.

#### 2.4.2. Clinical Learning Environment

We used the Clinical learning environment developed by Dunn and Burnett [19] and revised into Korean by Han [26]. This scale consists of 19 items with 5 sub-scales: staff–student relationship, hierarchy and ritual, nurse manager commitment, patient relationships, and student satisfaction. A higher score indicated higher-level adaptation in the clinical learning environment. For this study, a confirmatory factor analysis was performed to extract sub-factors. In Han’s [26] study, Cronbach’s α was 0.84, whereas, in this study, it was 0.85.

#### 2.4.3. Clinical Practicum Adaptation

We used a measurement scale developed by Yi [27] and used by Kim and Shin [28] to conduct a factor analysis of subcategories for nursing students. Fourteen items measured 3 sub-scales: adaptation to assignments, adaptation to the environment, and satisfaction with clinical practice, on a 5-point Likert scale. A higher score indicated higher-level adaptation in clinical practicum adaptation. For this study, a confirmatory factor analysis was performed to extract sub-factors. Yi [27] reported a Cronbach’s α of 0.86, Kim and Shin [28] reported Cronbach’s α = 0.85, and in this study, Cronbach’s α was 0.80.

#### 2.4.4. Nursing Professionalism

We measured nursing professionalism with a measurement scale developed by Yeun et al. [22] which was revised for nursing students by Han et al. [29]. Eighteen items measured 5 sub-scales: self-concept of the profession, social awareness, professionalism of nursing, the roles of nursing service and originality of nursing, rated on a 5-point Likert scale. A higher score indicated high-level nursing professionalism. For this study, a confirmatory factor analysis was performed to extract sub-factors. In Han et al.’s study [29], Cronbach’s α was 0.94, and in this study it was 0.84.

#### 2.4.5. Empathy

We measured empathy using the 28-item Interpersonal Reactivity Index (IRI) developed by Davis [30] and revised by Park [31]. All items were rated using a 5-point Likert scale. A higher score indicated high-level empathy. The tool received a Cronbach’s α ranging between 0.70 and 0.78 at the time of development [30], while this study it was 0.84.

#### 2.4.6. Person-Centered Care Competency

We measured PCCC using the 17-item scale developed by Suhonen et al. [32] and translated by Park [6]. The items measured 3 sub-scales: clinical situation, personal life situation, and decisional control, on a 5-point Likert scale. A higher score indicated high person-centered care competency. For this study, a confirmatory factor analysis was performed to extract sub-factors. In Suhonen et al.’s study [32], Cronbach’s α was 0.88, and in this study it was 0.84.

### 2.5. Data Analysis

We analyzed data using IBM SPSS^®^ version 26.0 and AMOS version 25.0, while the descriptive statistics used to analyze the respondent’s demographics, skills enhancement program scores, and clinical competence scores included mean, standard deviation, frequency, and percentage. The reliability of the questionnaire was verified using Cronbach’s coefficients. For sample normality, we obtained statistical data on skewness and kurtosis. Next, we subjected the model to a confirmatory factor analysis to affirm the measurement variable for the latent variable. For convergent sample validity, we used factor loadings, average variance extracted (AVE), and critical ratio (CR). We determined discriminant validity between variables using correlation coefficients (r) and AVE. We evaluated fitness for the measurement and hypothetical models using the χ^2^, χ^2^/df, adjusted goodness-of-fit index (AGFI), goodness-of-fit index (GFI), comparative fit index (CFI), root mean square error of approximation (RMSEA), and standardized root mean square residual (SRMR). To verify the statistical significance of the modified model, we used standard regression weights, critical ratio (CR), *p* values, and squared multiple correlation. The bootstrap maximum likelihood was used 5000 times within the 95% confidence interval to test the significance of the direct, indirect, and total effects of the modified model.

To verify the significance of the individual indirect effects on PCCC, we conducted an analysis using phantom variables. Regarding setting and analyzing phantom variables, a standardization coefficient cannot be identified; thus, we also verified the significance of the indirect effect on the final model through the non-standardized coefficient in this study [33].

## 3. Results

### 3.1. Participant Demographics

Of the 383 participants, 338 were female (88.25%) and 45 (11.75%) were male, including 151 (39.43%) and 232 (60.57%) third- and fourth-year students with an average age of 22.88 years (SD = 2.05). Most students (250: 65.27%) pursued a nursing career as their own choice, followed by those who did so due to other reasons (102: 26.63%), and, lastly, those who did so because of their grades (31:8.09%). Concerning a subjective health status, the vast majority of participants was very healthy (165; 43.09%), followed by healthy (132: 34.46%), and unhealthy (86: 22.45%) (Table 1).

### 3.2. Research Variables Normal Distribution

Table 2 shows the descriptive statistics of the measurement variables and multivariate normality. The latent variable factors’ multivariate normality was verified through standard deviations, skewness, and kurtosis; this study meets the criteria for the skewness and kurtosis values. The absolute value for the skewness of each variable did not exceed 1.97 and that of kurtosis did not exceed 2.58, based on Abidin and Brunner [34]; thus, every factor showed a normal distribution as shown in Table 2.

### 3.3. Confirmatory Factor Analysis of the Measurement Model

This study identified how well the measurement variables represented the latent variables through a confirmatory factor analysis. The standardization factors of the individual paths were shown to be 0.50 or higher and the CR^1^ (critical ratio) was at least 1.96. The AVE values ranged from 0.53 to 0.67 and CR^2^ (construct reliability) values ranged from 0.70 to 0.86, which satisfied the normal range: AVE value was 0.5 or higher and the CR^2^ (construct reliability) value was 0.7 or higher. This indicated that the measurement tool had a good convergent validity (Table 3).

### 3.4. Correlations between Variables

The correlations between the measurement variables were analyzed using Pearson’s correlation coefficient analysis (Table 4). We observed positive correlation among all individual variables. In this study, the correlation coefficient between latent variables was in the range of 0.2–0.6 and the squared value was smaller than every AVE value [35]; thus, discriminant validity was secured.

### 3.5. Testing the Structural Model

#### 3.5.1. Verification of Fit for the Hypothetical Models

The fitness indices of the theoretical model proposed in this study are shown in Table 5. The theoretical model was considered appropriate based on the following model fitness indices: CMIN/DF(χ^2^/df) = 2.73, GFI = 0.92, AGFI = 0.89, RSMSEA = 0.07, SRMR = 0.06, NNFI = 0.91. Therefore, the PCCC hypothesis model, which was established based on the goodness of χ², GFI, CFI, NFI, RMR, and RMSEA values, was considered appropriate (Table 5).

#### 3.5.2. Hypothesis Model Test and Effect Analysis

The standardization factors and CR values of the model were verified to determine whether there were direct or indirect relationships between the variables, as shown in Table 6. Figure 3 is a path diagram showing the influence pathways between the variables of the final PCCC model, considering the standardization factors affecting the study variables’ relationships.

Self-awareness had a direct effect on PCCC (β = 0.53, *p* < 0.001, and the clinical learning environment had a direct effect on PCCC (β = 0.50, *p* < 0.001). Clinical practicum adaption had a direct effect on nursing professionalism (β = 0.59, *p* < 0.001). Self-awareness and the clinical learning environment had indirect effects on nursing professionalism (β = 0.30, *p* < 0.001; β = 0.31, *p* < 0.001).

Additionally, self-awareness and the clinical learning environment had direct effects on empathy (β = 0.36, *p* < 0.001; β = 0.17, *p* < 0.001).

Nursing professionalism and empathy had direct effects on PCCC (β = 0.49, *p* < 0.001; β = 0.31, *p* < 0.001). Additionally, self-awareness, the clinical learning environment, and clinical practicum adaption had indirect effects on PCCC (β = 0.26, *p* < 0.001; β = 0.20, *p* < 0.001; β = 0.29, *p* < 0.001).

Ultimately, PCCC was influenced by the following variables in descending order: nursing professionalism, empathy, clinical practicum adaptation, self-awareness, and the clinical learning environment. The explanatory power of these variables was 38.8%. All paths in the modified model were statistically significant (Figure 3 and Table 6).

#### 3.5.3. Mediating Effect Analysis by Phantom Variable

We verified the significance of the individual mediating effects between self-awareness and PCCC, the clinical learning environment and PCCC by setting phantom variables. From the mediating effect analysis, the significance test showed the following: self-awareness had an indirect effect on PCCC through clinical practicum adaptation and nursing professionalism (B = 0.21, *p* < 0.001). Self-awareness also had an indirect effect on PCCC through empathy (B =0.15, *p* < 0.001). In addition, the clinical learning environment had indirect effects on PCCC through clinical practicum adaptation and nursing professionalism (B = 0.12, *p* < 0.001). The clinical learning environment had an indirect effect on PCCC through empathy (B = 0.04, *p* < 0.001; Table 7).

## 4. Discussion

We aimed to construct a model that would empirically show a structural relationship between nursing students’ self-awareness, the clinical learning environment, clinical practicum adaption, nursing professionalism, empathy, and PCCC by verifying an explanatory hypothetical model based on the SCCT.

We used the SCCT and previous related findings to identify the factors affecting PCCC among senior nursing students, thereby constructing and verifying an explanatory hypothetical model (related factors showed an explanatory power of 38.8%). Model fitness was also high; thus, indicating suitability for predicting PCCC. Specifically, self-awareness and the clinical learning environment affected the learning experience and expectation factors (clinical practicum adaptation and nursing professionalism) and empathy, while nursing professionalism and empathy ultimately influenced PCCC. These results support the SCCT, in that personal and contextual factors affected expectation factors through learning experience, which ultimately affected goal-related behavior.

Of the results, nursing professionalism was the most influential variable that significantly affected PCCC; thus, supporting Lee et al. [36] and Oh’s [37] studies, which emphasized that nursing professionalism was the most significant factor for developing PCCC.

Medical and social changes due to increasing educational levels and the demands of subjects, caused by recent developments in medical technology, require nurses to make decisions in more diverse and complex ethical situations. Given that Kaya and Dalgiç [38] indicated that the key attribute of nursing professionalism is high ethical standards, enhancing nursing students’ ethical sensitivity and standards can be a stepping-stone for nurses to apply PCC.

Thus, nursing students need to set their own nursing professionalism directions from the beginning of their undergraduate programs. In addition, there is a need to create a culture in which undergraduates can take interest in developing nursing professionalism. For example, this could be performed by providing an opportunity to select activities, such as nursing association activities autonomously, as it is necessary to approach the role of nurses from a macro perspective. In addition, nursing instructors could play the role of a facilitator so that, through the development of various curriculum and non-curricular programs/activities, such as “introductory nursing studies” or “nursing professionalism courses”, appropriate nursing professionalism can be formed.

The second variable that influenced PCCC was empathy; thus, supporting previous studies that show that empathy is not only a key factor for PCCC, but also that the two are highly correlated [10,39,40]. Derksen et al. [40] also found that empathy could be used to build supportive relationships while enhancing patient comfort. As nurses must play key roles in determining the needs of patients through individualized approaches rather than solely focusing on acute treatments, this requires continual practical revisions of the nursing curriculum to ensure that empathy is highlighted and properly developed. Thus, improvements may include adding curriculum updates, related teaching strategies (problem-based learning), role-playing, and high-fidelity simulations based on understanding and communicating with others [41].

The third variable affecting PCCC was clinical practicum adaptation; thus, supporting Haugland and Giske [42], who emphasized that clinical practice experience was the most important variable for developing PCCC. Combined with practical education, this shows that experience can instill person-centered values/attitudes and nursing knowledge [27,43]. Thus, successful adaptation to clinical practice should be a significant factor for achieving PCCC among nursing students who establish their roles through appropriate interactions with patients. As clinical practice experience serves as a preparatory stage for patient care by allowing nursing students to internalize their beliefs, values, and ethical standards, it is important to ensure that a range of emotional and social aspects are incorporated into clinical practice education and a wide range of actual clinical settings [3,42].

Self-awareness was the fourth factor that influenced PCCC, which Haley et al. [10] and Ayed et al [44] pointed out as a significant prerequisite for providing PCCC. Our results also supported those of Im [45], who argued that interpersonal understanding was enhanced through the proper recognition of oneself.

Furthermore, nursing students’ self-awareness was found to be a predictor of PCCC mediated by clinical practicum adaptation and nursing professionalism. This coincided with previous studies reporting that through deep self-reflection and intrinsic motivation, high-level nursing professionalism can be formed [10,18].

Moreover, in another pathway, self-awareness predicted PCCC through empathy. This was consistent with previous results stating that people with high self-awareness have high empathy for others; the attribute of recognizing and feeling others’ emotions (empathy) has some similarities with the attribute of self-awareness [10].

Thus, self-awareness intervention in the nursing curriculum could further enhance interpersonal understanding, thereby increasing PCCC [4,10]. In other words, effective PCCC can be achieved through emphasizing clear personal beliefs and values while providing students with opportunities to learn about themselves. To increase these traits in nursing students, what may be required is that each students’ level of self-awareness is understood from the time of admission and that they are systematically led to developing empathy through mid- to long-term plans at the individual level until graduation. For example, this could be performed by encouraging nursing students to focus on their own thoughts and feelings and to increase self-awareness through activities, such as meditation, prayer, yoga, and mindfulness

The clinical learning environment was the last factor affecting PCCC; thus, supporting previous studies that showed a significant correlation between PCCC and the environment [46,47,48]. We, thus, infer that supportive cultures should be maintained through the establishment of mutually cooperative relationships among many interacting clinical professionals, such as clinical practice professors, practice instructors, and heads of institutions.

In addition, the nursing students’ clinical practice environment was found to be a predictor of PCCC mediated by nursing professionalism and clinical practicum adaptation. Currently, clinical nursing education sites are striving to maintain a certain level of quality; however, it is difficult to maintain consistency in all sites due to various reasons, such as the participants’ severity, nurse personnel, and field conditions. Here, we can consider that our result were a consequence of clinical experiences—through senior nurses in different fields, field leaders’ leadership types, and clinical practice facilities affecting the formation of nursing professionalism—ultimately affecting PCCC [49,50]. Based on these results, the following efforts are required. First, instructors should provide nursing students with sufficient information about their clinical practice before undertaking clinical practice to improve their adaptation to the clinical site. Secondly, maintaining cooperation with field leaders to create an educational atmosphere based on role models who demonstrate expertise and skills at the clinical practice department level should be continued. Finally, cooperative relationships between schools and practice hospitals at the institutional level should be maintained.

Finally, the nursing students’ clinical practice environment was found to be a predictor of PCCC mediated by empathy. Recently, various difficulties have arisen in nursing students’ clinical practice education due to COVID-19. From this, we can judge that the implications of this study are crucial. Due to COVID 19, opportunities for the improvements of interpersonal relationships have been gradually decreasing. Therefore, by enhancing nursing students’ empathy through communication programs and simulation-based scenarios that require empathy for the patients, the limited clinical education can be supplemented, and a better planned, PCCC-promoting, and systematic nursing education system for nursing students can be realized without restrictions on social change.

In this study, the influence of the variables was deemed appropriate, accounting for 38.8% of PCCC. Therefore, the variables suggested in this study can be considered influencing factors when developing a PCCC-enhancing program.

This study is meaningful, in that it systematically and comprehensively presented the multidimensional factors affecting the PCCC of senior nursing students by applying the SCCT model, while previous studies have confirmed the influence of sociodemographic factors, psychological factors, and relational factors.

## 5. Conclusions

This study established and verified a theoretical framework for the comprehensive understanding of senior nursing students’ PCCC based on the SCCT. The final model established through this study was confirmed to be suitable for explaining and predicting PCCC. Here, nursing professionalism had the greatest influence on the PCCC for senior nursing students, followed by self-awareness, empathy, clinical practicum adaptation, and the clinical learning environment, explaining 38.8% of PCCC.

It is believed that the effort to identify and enhance the PCCC of nursing students is of great importance as students grow to become healthcare professionals working in various environments. Ultimately, these efforts will help nursing students become nurses who provide holistic care by respecting the beliefs and values of their patients.

Based on this study, we suggest that, first, a program to improve nursing students’ PCCC based on the variables identified through this study be established and the effects of these programs be identified, and second, the PCCC levels be measured continuously after graduation.

## Figures and Tables

**Figure 1 ijerph-18-10421-f001:**
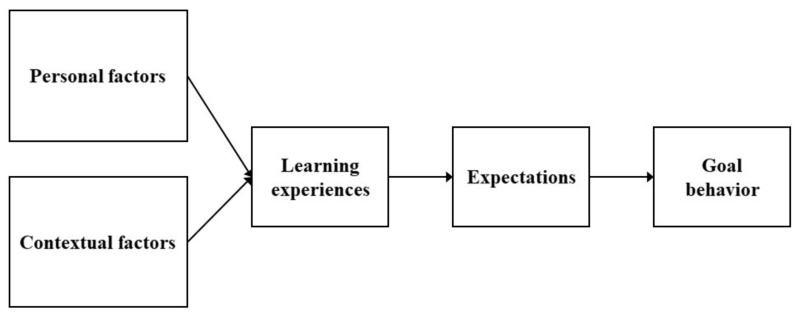
The conceptual framework of this study, based on the social cognitive career theory by Lent, Brown, and Hackett [17].

**Figure 2 ijerph-18-10421-f002:**
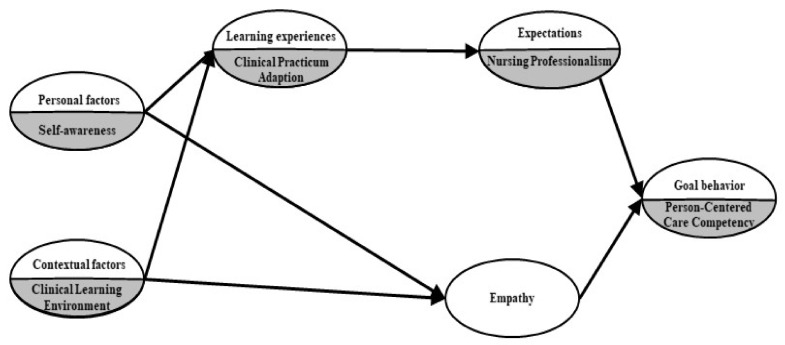
The hypothetical model of this study, based on the social cognitive career theory by Lent, Brown, and Hackett [17].

**Figure 3 ijerph-18-10421-f003:**
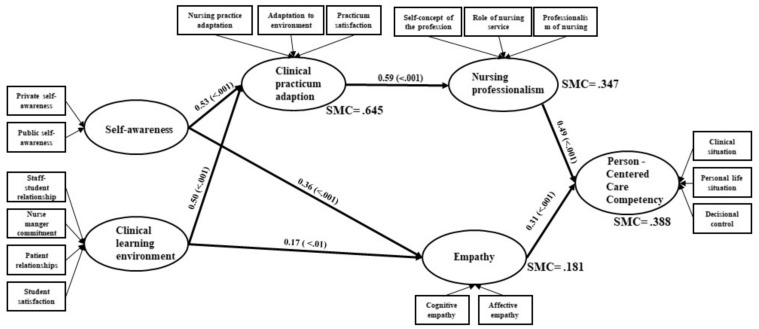
Path diagram of the modified model: Person-centered care competency of nursing students (Standardized regressing coefficients); SMC —squared multiple correlations.

**Table 1 ijerph-18-10421-t001:** Characteristics of participants (*N* = 383).

Characteristics/Classifications	M ± SD or *N*	%
Age (y)	22.88 ± 2.05	
Academic year		
Third year	151	39.43%
Fourth year	232	60.57%
Gender		
Female	338	88.25%
Male	45	11.75%
How tuition is paid		
With the help of parents	232	60.57%
Independently	151	39.43%
Religion		
Have	134	34.99%
None	249	65.01%
Hospitalization experience		
Experience	180	47.00%
No Experience	203	53.00%
Experience in caring for inpatients		
Experience	157	40.99%
No Experience	226	59.01%
Location of university		
Central Province	79	20.63%
Honam Province	184	48.04%
Yeongnam Province	120	31.33%
Motivation for applying to nursing		
Decided by myself	250	65.28%
Applied according to grades	31	8.09%
Other reasons	102	26.63%
Subjective health status		
Very healthy	165	43.09%
Healthy	132	34.46%
Unhealthy	86	22.45%

**Table 2 ijerph-18-10421-t002:** Person-centered care competency related variables (*N* = 383).

Latent Variable	Observed Variable	Mean	SD	Skewness	Kurtosis
Self-awareness	Private self-awareness	3.49	0.44	−0.21	−0.06
	Public self-awareness	3.34	0.43	−0.21	−0.06
Clinical Learning Environment	Staff–student relationship	3.24	0.79	−0.21	−0.06
	Nurse manger commitment	2.93	0.59	−0.03	0.18
	Patient relationships	3.18	0.75	−0.19	−0.03
	Student satisfaction	3.18	0.66	−0.24	0.50
Clinical Practicum Adaption	Nursing practice adaptation	3.54	0.69	−0.44	0.41
	Adaptation to environment	3.30	0.68	−0.03	0.11
	Practicum satisfaction	3.38	0.60	0.34	0.51
Nursing Professionalism	Self-concept of the profession	3.89	0.60	−0.38	0.05
	Role of nursing service	4.12	0.75	−1.14	1.37
	Professionalism of nursing	3.94	0.79	−0.69	0.87
Empathy	Cognitive empathy	3.63	0.50	−0.11	0.42
	Affective empathy	3.37	0.44	−0.21	0.85
Person-Centered Care Competency	Clinical situation	3.95	0.57	−0.14	−0.08
	Personal life situation	3.68	0.72	−0.02	−0.28
	Decisional control	3.91	0.60	0.01	−0.68

**Table 3 ijerph-18-10421-t003:** Confirmatory factor analysis of the measurement model.

Latent Variable	Observed Variable	B	β	SE	CR^1^	AVE	CR^2^
Self-awareness						0.53	0.70
→	Public self-awareness	1.00	0.79				
	Private self-awareness	0.82	0.67	0.09	8.97 ***		
Clinical Learning Environment						0.61	0.86
→	Staff –student relationship	1.00	0.70				
	Nurse manger commitment	0.75	0.70	0.06	12.70 ***		
	Patient relationships	1.21	0.90	0.08	15.41 ***		
	Student satisfaction	0.94	0.79	0.07	14.16 ***		
Clinical Practicum Adaption						0.57	0.80
→	Nursing practice adaptation	1.00	0.77				
	Adaptation to environment	0.99	0.78	0.07	14.15 ***		
	Practicum satisfaction	0.82	0.72	0.06	13.21 ***		
Nursing Professionalism						0.65	0.85
→	Self-concept of the profession	1.00	0.73				
	Role of nursing service	1.38	0.81	0.09	14.76 ***		
	Professionalism of nursing	1.60	0.88	0.10	15.50 ***		
Empathy						0.50	0.70
→	Cognitive empathy	1.00	0.88				
	Affective empathy	0.55	0.58	0.08	7.25 ***		
Person-centered care competency						0.67	0.86
→	Clinical situation	1.00	0.85				
	Personal life situation	1.09	0.88	0.06	18.86 ***		
	Decisional control	1.00	0.68	0.07	14.10 ***		

*** *p* < 0.001; CR^1^—critical ratio; AVE—average variance extracted; CR^2^— construct reliability.

**Table 4 ijerph-18-10421-t004:** Correlations between variables.

Latent Variables	1	2	3	4	5	6
1. Self-awareness	1					
2. Clinical Learning Environment	0.20	1				
3. Clinical Practicum Adaption	0.60	0.60	1			
4. Nursing Professionalism	0.42	0.40	0.52	1		
5. Empathy	0.42	0.25	0.31	0.46	1	
6. Person-Centered Care Competency	0.50	0.31	0.58	0.58	0.53	1

**Table 5 ijerph-18-10421-t005:** Model fit.

Fitting Index	CMIN/DF	GFI	AGFI	RMSEA	CFI	SRMR	NNFI
Result	2.73	0.92	0.89	0.07	0.94	0.06	0.91
Criteria	<3	>0.9	>0.8	<0.08	>0.9	<0.05	>0.9

CMIN/DF — chi-square fit statistics/degree of freedom; GFI— goodness of fit index; AGFI — adjusted goodness of fit index; RMSEA — root mean square error of approximation; CFI — comparative fit index; SRMR — standardized root mean square residual; NNFI —non normed fit index

**Table 6 ijerph-18-10421-t006:** Path analysis between variables of the study model.

Direction		B	β	SE	CR	Direct Effect	Indirect Effect	Gross Effect	SMC
CPA									0.645
←	SA	0.83	0.53	0.11	7.68 ***	0.53 ***		0.53 ***	
←	CLE	0.48	0.50	0.06	8.52 ***	0.50 ***		0.50 ***	
NP									0.347
←	CPA	0.49	0.59	0.05	8.99 ***	0.59 ***		0.59 ***	
←	SA						0.30 ***	0.30 ***	
←	CLE						0.31 ***	0.31 ***	
Empathy									0.181
←	SA	0.58	0.36	0.10	5.74 ***	0.36 ***		0.36 ***	
←	CLE	0.17	0.17	0.05	3.15 **	0.17 **		0.17 **	
PCCC									0.388
←	NP	0.51	0.49	0.06	8.22 ***	0.49 ***		0.49 ***	
←	Empathy	0.26	0.31	0.06	4.51 ***	0.31 ***		0.31 ***	
←	SA						0.26 ***	0.26 ***	
←	CLE						0.20 ***	0.20 ***	
←	CPA						0.29 ***	0.29 ***	

**** p* < 0.001; *** p* < 0.01; CR—critical ratio; SE—standard error; SMC—squared multiple correlations; CPA—clinical Practicum adaption; SA—self-awareness; CLE—clinical learning environment; NP—nursing professionalism; PCCC—person-centered care competency.

**Table 7 ijerph-18-10421-t007:** Mediating effect analysis by phantom variable.

Direction							Direct Effect (B)	Indirect Effect (B)	Gross Effect (B)
Self-awareness	→	Clinical Practicum Adaption	→	Nursing Professional Value	→	Person-Centered Care Competency	-	0.21 ***	0.21 ***
Self-awareness	→	Empathy	→			Person-Centered Care Competency	-	0.15 ***	0.15 ***
Clinical Learning Environment	→	Clinical Practicum Adaption	→	Nursing Professional Value	→	Person-Centered Care Competency	-	0.12 ***	0.12 ***
Clinical Learning Environment	→	Empathy	→			Person-Centered Care Competency	-	0.04 **	0.04 **

*** *p* < 0.001; ** *p*< 0.01.

## Data Availability

The data that support the findings of this study are available from the corresponding author upon reasonable request.

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
