# Peer review of "Structural Equation Model for Developing Person-Centered Care Competency among Senior Nursing Students"

_ijerph, 2021, doi:10.3390/ijerph181910421_

Round 1

Reviewer 1 Report

Dear Authors,

Thanks for the opportunity to review this paper. The theme is extremely relevant for nursing education and for the quality of the health care provided by nurses. The introduction is well-grounded, although the SCCT, should be more detailed in order to improve clarity (especially for readers who are not familiar with the theory).

Regarding figure 2, authors should specify what they mean when stating that "Figure 2 shows the influencing relationships between the variables of the final study model, considering non-standardization and standardization factors of the relationships between the study variables". How they achieved the Indirect Effects and the Gross Effects presented in table 7.  Regarding table 7 and figure 2, authors should indicate if the values presented are standardized or non-standardized. 

It was a pleasure to read this paper, and I hope my comments can contribute to improve the quality of the manuscript.

Author Response

Authors’ response

We thank you for your thoughtful suggestions and insights, which have further enriched our manuscript and produced a better and more balanced account of the research. We have rechecked the manuscript and made appropriate changes in accordance with your suggestions.

We have also tried our best to make the revisions as sincere as possible.

Our responses to your comments are included herewith.

In addition to the revisions described below, please let us know if the manuscript can benefit from any more edits, and we will do our best to incorporate them as needed.

The revised parts are highlighted in red.

Thank you in advance.

Sincerely,

Reviewer 2 Report

I would like to thank the editor for offering this opportunity to review this manuscript. The purpose of this study is “to verify the suitability of the person-centered care competency (PCCC) model for senior nursing students based on the Social Cognitive Career Theory (SCCT); and to clarify the direct and indirect effects of related factors on PCCC”. This is an important topic and findings of the study will make contribution in strategies and interventions designed to enhance person-centered care competency for senior nursing students. The manuscript is generally well written; reflects rigorous methodology. My comments to enhance the quality of the manuscript are few.

  1. Page 3, 1.2. Conceptual Framework and Hypothetical Model: it is difficult to follow or figure out what is the component included in SCCT, and what is the conceptual framework of this study.
  2. Page 7, Table 1: clarification is needed for subjective health status, particularly for those of unhealthy students (86, 22.45%).
  3. Page 8, the first paragraph: two abbreviations of CR are confusing.

Author Response

(The authors gave the same response as above.)
